# European Forest Governance: Status Quo and Optimising Options with Regard to the Paris Climate Target

**Jessica Stubenrauch [1], Beatrice Garske [2,3], Felix Ekardt [2,4,*] and Katharina Hagemann [2]**

[1] Helmholtz Centre for Environmental Research, 04318 Leipzig, Germany; jessica.stubenrauch@ufz.de
[2] Research Unit Sustainability and Climate Policy, 04229 Leipzig, Germany; beatrice.garske@uni-rostock.de (B.G.); katharina.hagemann@posteo.de (K.H.)
[3] Faculty of Agricultural and Environmental Sciences, University of Rostock, 18051 Rostock, Germany
[4] Faculty of Law and Interdisciplinary Faculty, University of Rostock, 18051 Rostock, Germany
[*] Correspondence: felix.ekardt@uni-rostock.de

**Abstract:** This article assesses and develops policy instruments for forest governance in the EU. Methodologically, it examines opportunities and limits for negative emissions by means of a literature review. On this basis, it conducts a qualitative governance analysis of the most important instruments of EU forest policy and presents optimizing policy options, measured against the binding climate and biodiversity targets under international law. Our analysis shows that the potential benefits of afforestation and reforestation for climate mitigation are overestimated, and are often presented as the new saviours to assist in reaching climate neutrality, inter alia, since only biodiverse and thus resilient forests can function as a carbon sink in the long term. Furthermore, we demonstrate that the existing EU law fails to comply with climate and biodiversity targets. Quantity governance systems for livestock farming, fossil fuels and similar drivers of deforestation represent a more promising approach to forest governance than the dominant regulatory and subsidy-based governance. They are most effective when not directly addressing forests due to their heterogeneity but central damaging factors such as fossil fuels and livestock farming. Selected aspects of regulatory and subsidy law can supplement these quantity governance systems when focusing on certain easily attainable and thus controllable subjects. These include, e.g., the regulatory protection of old-growth forests with almost no exceptions and a complete conversion of all agricultural and forest subsidies to "public money for public services" to promote nature conservation and afforestation.

**Keywords:** forest; governance; climate; biodiversity; Paris Agreement; EU law; international law

## 1. Problem Statement and Research Issue

The future development of land use, land-use change and forestry (LULUCF) sector is of crucial importance for combating climate change and the long-term preservation of natural resources as well as protecting biological diversity [1–3]. This is especially the case for the overall land sector, including agriculture in general, forestry and other land use (AFOLU). From a climate perspective, the unique characteristic of the sectors is that they do not only account for greenhouse gas (GHG) emissions but also serve as a sink for GHGs. There is an enormous potential for natural carbon storage by soils and the upstanding biomass, particularly forest ecosystems, peatlands and other wetlands as well as arable land, provided these environmental compartments remain intact or are restored and used in a sustainable way, preserving natural functions [2,4–8]. It must be noted, however, that the international law term LULUCF does, in contrast to AFOLU, not cover some core sectors connected to land use that represent high emission levels—namely livestock farming and fertiliser production [9,10].

The exact strategy for forests (and negative emission options in general) is always dependent on the targets that must be fulfilled. According to Art. 2 para. 1 of the Paris

Agreement (PA) [11], global warming should be limited to well below 2 °C compared to pre-industrial levels and efforts should be pursued to stay within a 1.5 °C temperature limit. To reach greenhouse gas neutrality, zero fossil fuels and a massive reduction of livestock farming are necessary but not sufficient (see in detail [1,12,13]). In the future, all inevitably occurring GHG have to be compensated for by the creation of negative emissions in sinks [1,14–16]. The exact amount of negative emissions needed is still an open question as well as how they can be generated. This always depends on the efforts to cut down GHG emissions.

In this context, alongside enhanced soil carbon sequestration in agriculture [17–19], reforestation, forest restoration and large-scale afforestation are increasingly discussed in IPCC climate scenarios as nature-based negative emission technologies (NETs) [14,20]. Bastin et al., estimate that 1 billion hectares globally are available for additional forest without using agricultural or urban land. This could contribute to limiting global warming to 1.5 °C by 2050 [14,21]. However, there is a lively scientific debate on the degree to which forests and natural sinks, in general, can or have to contribute to climate protection or whether large-scale technical approaches in the field of geoengineering have to be considered as well [14,21–25]. Most geoengineering techniques are thus still in development and might pose additional threats to human rights, while their effectiveness in climate protection remains largely unproven [12,14,16]. In contrast, natural sinks like forest ecosystems already play an important role in stabilising the climate [7].

In earlier analyses, we have taken a closer look at peatlands that bear the promise of combining negative GHG emissions with biodiversity protection [4]—and problematic technological approaches to negative emissions called geoengineering [12]. In the present contribution, we will focus on a critical assessment of the potential of forests ecosystems in climate as well as for biodiversity protection [12,14,26], also considering the manifold interactions with other types of land use. On this basis, we will assess the status quo of EU forest governance and develop policy-optimizing options. In our earlier studies, some problems in governing the land-use sector have been identified, especially the problem of depicting climate and biodiversity effects in highly heterogeneous landscapes and the major role of addressing fossil fuels and livestock farming as damaging factors for finding integrated solutions to various environmental challenges. These problems and aspects will also play a major role in the present study, which will, by these means, contribute a new dimension to the overall discussion in sustainability governance on various policy instruments such as regulatory law, subsidies, levies, and cap-and-trade schemes.

## 2. Methodology, Taking Environmental Targets and Governance Problems into Account

As a first step, the article critically reviews the literature on the natural scientific debate on forest ecosystems and their potential contribution to climate protection depending on the type of forests, their different phases of growth and varying climatic conditions, including the maximum sink capacity to be achieved by reforestation, afforestation, or the preservation of old or primary forest ecosystems. Building on this, a multi-methodological qualitative governance analysis (or steering analysis) will be applied to assess the effectiveness of existing policy instruments and potential future policy instruments regarding forests and land use [4,27]. The effectiveness of existing and potential policy instruments is (on the basis of understanding the natural environment) measured against (a) normative standards given by political targets, (b) the ability to avoid typically recurring governance problems (such as the above-mentioned problem of depicting, rebound effects, geographical or sectoral shifting effects and enforcement deficits), and (c) incorporates knowledge from different scientific backgrounds like natural science and human behaviour (see Figure 1). This methodology has been described several times in more detail in the present journal and elsewhere since 2018 [4,9,10,12,27,28].

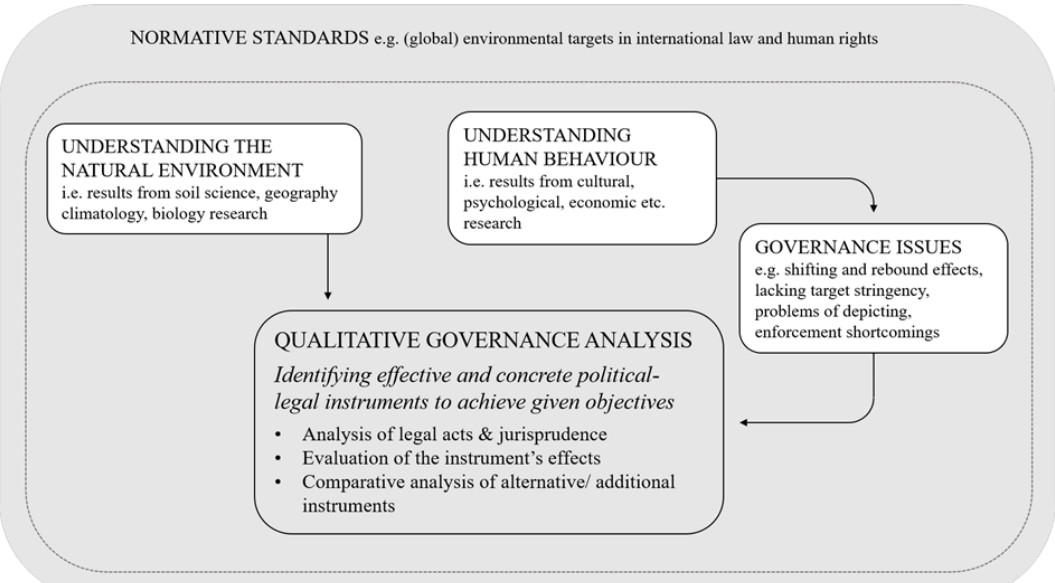

**Figure 1.** Elements of a qualitative governance analysis [28].

As mentioned in Section 1, according to Art. 2 para. 1 of the Paris Agreement, global warming should be limited to well below 2 °C compared to pre-industrial levels, and efforts should be pursued to stay within a 1.5 °C temperature limit. We have shown elsewhere [12,15,27] that this contains a legally binding obligation to try to stay within the 1.5 °C limits (the binding character and the focus on 1.5, not 2 degrees, is also adopted by the German Federal Constitutional Court, Order of 24 March 2021, 1 BvR 2656/18 et al.). To meet this limit with a probability of clearly more than 67% (since 50–67% is not enough from the legal point of view; see [12,15,27]) and given equal per-capita emission rights on a worldwide scale, globally net-zero emissions across all sectors are required within a probable maximum of less than two decades, likely before 2035 (shown by [15] discussing limitations of [14], [1,13]). The question whether it is sufficient to achieve climate neutrality only in the second half of this century to meet the legally binding temperature target of Art. 2 para. 1 PA, declared as a long-term temperature goal in Art. 4 para. 1 PA was answered in an earlier contribution [15] in favour of Art. 2 para. 1 PA. The authors conclude that compliance with the 1.5 °C limit needs to be reached much earlier with a high probability and without an overshoot of temperature [12,15,29].

Furthermore, the CBD aims at halting global biodiversity loss. The subsequent EU target for 2020, as a normative basis of the 2011 biodiversity strategy, was to halt the loss of biodiversity and degradation of ecosystem services in the EU and restore them as far as possible while increasing the EU's contribution to preventing biodiversity loss globally. However, it has repeatedly become clear that this target is being missed by a wide margin [30–32]. As a follow-up, the Kunming Declaration [33] was announced on 13 October 2021, aiming to establish a post-2020 global biodiversity framework regarding biodiversity loss [34]. According to the declaration, inter alia, biodiversity protection should be considered across all legal decision-making processes, harmful subsidies should be phased out and redirected and the rights of Indigenous people should (finally) be protected in the future. Apart from that, the call to protect and conserve 30% of terrestrial and marine areas "through well-connected systems of protected areas and other effective area-based conservation measures by 2030" (Kunming Declaration, 3) is noted. However, thus far, the rather "vague commitments that lack accountability are hardly a step forward from the 2010 Aichi targets" [35]. In the end, the outcomes of the negotiation processes following the vision of "Living in Harmony with Nature" in 2050 will be decisive as to how far the colossally missed biodiversity targets from the CBD can be achieved in the future.

It generally must be considered that biodiversity is difficult to measure and therefore difficult to translate into an operationalizable ecological target. Ultimately, limiting global warming is easier to operationalize (via a GHG emission cap) than protecting biodiversity or restoring ecosystems. Nevertheless, it makes sense to consider the CBD as a complement to the Paris Agreement because climate change is closely intertwined with other sustainability issues like biodiversity loss but also disrupted nitrogen (N) and phosphorus (P) cycles, and water and soil pollution [27,36,37].

In contrast, a "natural forest condition" (occasionally cited) would be unsuitable as a guiding star from the outset. In view of the processuality of eco-systemic events, this can be understood to mean completely different states and points in time in natural history since the last ice age. Given this, the implication of terms such as "natural" or "close to nature" is untenable that it can be decided scientifically which treatment of the forest is to be aimed for. Rather, this is an ethical, legal and political question [27].

## 3. Results: Potential and Limits of Forest Ecosystems on Climate and Biodiversity Protection and Implications for the Legislative Process

### 3.1. The Importance of and Risks for Existing Forest Ecosystems

For decades, the world's forests have faced accelerating degradation and loss, impairing nature's balance, biodiversity and climate protection to a potentially life-threatening extent [2,7]. On the one hand, the irretrievable loss of flora and fauna is weakening functioning ecosystems as the basis of all life on earth [26,38,39]. On the other hand, the sink capacity for GHG emissions—needed more urgently than ever in human history to fight the climate crisis—is steadily decreasing [40]. Since 1990, approximately 420 million hectares of forest have been lost due to their conversion to other land uses [40]. Primary forests, the lungs of the earth, decreased by over 80 million hectares during that time [40]. The development is, therefore, already close to passing irreversible tipping points (on the example of the Amazon, see [41–44]).

One of the main reasons for the ongoing land-use changes causing deforestation is an agricultural expansion for the production of animal food (cattle ranging, soybean production). Other causes include palm oil production and various implications of the use of fossil fuels such as growing cities, expanding road construction, etc. [10,27,40,45–47]. Today, approximately 70–85% of the world's farmland is dedicated to animal-derived food production, such as meat and dairy products [10,48]. This shows a tremendous impact on both the occurring GHG emissions from the LULUCF sector and the globally accelerating biodiversity loss due to increased land-use pressure [10,48,49]. Concerning climate protection, there are estimates that, assuming a no-animal scenario, 6.6 gigatons $CO_{2equ}$ per year could be avoided, corresponding to 49% of the total GHG emissions of the food sector, and the sink capacity of terrestrial ecosystems could be enhanced by 8.1 gigatons $CO_2$ on average each year in a 100-year time span [48]. While livestock farming, for several reasons, could (and should) be drastically reduced but cannot be cut back to zero worldwide as we have discussed elsewhere [9,10,15,27,36], it nevertheless becomes clear that drastically minimised livestock farming and a phasing-out of fossil fuels are indispensable to combat the climate and biodiversity crisis and to protect and/or restore worldwide forests [1,10,13,16,27,50,51]. In addition to carbon dioxide emissions ($CO_2$), livestock farming is one of the primary contributors to non-$CO_2$ emissions such as methane ($CH_4$), nitrous oxide ($N_2O$), and nitrogen oxides ($NO_x$) [52,53].

Today, not least due to the transition to a post-fossil society, forests worldwide are under unprecedented pressure of use and are exposed to changing climatic conditions, threatening the existence of the last primary forests in particular. Thus, in the future, policy instruments will need to be designed to interact in a way to halt the globally accelerating decline of forests and either strictly protect remaining primary, old-growth and species-rich natural forests, following the principle of segregation, or ensure a sustainable

and multifunctional forest use in clear favour of biodiverse forest ecosystems. Therefore, forest cover worldwide needs to be mapped and monitored more sufficiently [54,55].

Considering the problem of deforestation, it is important to highlight that the demand-side is neither locally nor globally fixed but is determined by consumption patterns which can (and have to be) changed by effective policy instruments. Thus, a strong focus needs to be set on demand-sight climate mitigation measures to minimise land-use pressures in favour of intact forests, tackling the livestock farming and the biomass sectors in particular [10,20,27,56]. However, the implementation of policy instruments thus far addressing the demand-sight and thus, the drivers of forest loss—as we will show in the governance analysis in Section 4 in more detail—are also widely missing resulting in the acceleration of direct and indirect land-use changes [57–59].

Concerning the use of biomass, not only the construction, textile or chemical sectors, but also the substitution of fossil-fuel based plastics might lead to a higher demand for timber in the future [36,60]. It is therefore prudent to foster the reuse of resources, enhanced recycling and the cascade utilisation of wood. Forest governance must be integrated into a concept of circular economy, including efficiency, consistency and frugality strategies [27,36,61–63]. The latter is even relevant when deadwood or agricultural waste is used for energy purposes. Coarse, woody debris releases carbon more slowly and is more compliant with the natural carbon cycle than if energetically used [20,64] and agricultural wastes are important organic fertilisers that can contribute to the substitution of mineral fertilisers in the future [20,36,37].

To guarantee the protection and the reconciliation of both climate and biodiversity, it is crucial to avoid conflicting goals and use synergies. This is also essential for facilitating health provisions by forests, as reforestation and afforestation in the form of plantations can, next to forest clearance, be responsible for outbreaks of infectious diseases [65]. We have already seen that reducing land-use pressure caused by fossil fuels and animal husbandry could be a key element for this. Furthermore, reducing the usage of land-based biomass might therefore bear immense potential to reduce $CO_2$ emissions and decrease land-use pressures at the same time [20].

### 3.2. A Critical Review of Natural Scientific Data on Forests in the Climate Discourse and Implications for the Legislative Process

The carbon storage potential of forests is increasingly stressed within the climate mitigation debate. Thus, the present chapter seeks to answer two main questions with major significance regarding the assessment and development of policy instruments: Firstly, which contribution to climate (and biodiversity) protection can be expected to be provided by the forest sector and particularly afforestation projects in the future, and secondly, can this contribution be reliably measured against a specific baseline?

### 3.2.1. Carbon Sequestration Potential of Forests

Forest ecosystems contribute to approximately 50% of terrestrial net primary production and store approximately 45% of total terrestrial carbon and are therefore a crucial element in the global carbon cycle [66,67]. Forest biomass becomes a carbon sink as soon as the biological $CO_2$ uptake is higher than the total release of GHGs (e.g., through respiration, forest fire, profound disturbances; see [68]). The net carbon balance of forest ecosystems is regularly positive and even old-growth forests are not carbon-neutral per se, [69,70] and are able to further sequester carbon [54,71,72].

The carbon sequestration rate of forests depends on the type, age and density of trees, soil properties as well as latitude and connected climatic influences (e.g., temperature, precipitation, $CO_2$ concentration, nitrogen (N) deposition, and ozone ($O_3$) exposure) [54,73–76]. With increasing latitude, the potential of forests to store carbon generally decreases due to reduced net productivity [77]. Tropical forests have the largest potential to store carbon and also function as biodiversity hotspots.

The total carbon storage in forest ecosystems consists of carbon sequestered in the forest biomass (including stem biomass, coarse woody debris, roots) and in the soil organic matter (SOM) [54,70,74]. Soils store most of the total carbon in forest ecosystems [54,67,73]. The amount of carbon sequestered in forest soils depends on their specific characteristics, which in turn are influenced by the upstanding trees and their productivity. Luyssaert et al., 2008 estimate for old-growth forests older than 200 years that they sequester $2.4 \pm 0.8$ tons of carbon per hectare and year (t C ha$^{-1}$ yr$^{-1}$) on average, thereof $0.4 \pm 0.1$ t C ha$^{-1}$ yr$^{-1}$ in the stem biomass, $0.7 \pm 0.2$ t C ha$^{-1}$ yr$^{-1}$ in the coarse, woody debris (deadwood) and $1.3 \pm 0.8$ t C ha$^{-1}$ yr$^{-1}$ in the roots and the SOM.

Degradation processes or unsustainable forest management might further harm the carbon stock of forest ecosystems. This is why the sink capacity of forests is regularly overestimated. According to Tubiello et al. [78], the net contribution of worldwide forests for the period 2011–2020 was calculated to be less than $-0.2$ Gt $CO_2$ yr$^{-1}$, when net forest conversion emissions (3.1 Gt $CO_2$ yr$^{-1}$) were offset with net removals from forest land ($-3.3$ Gt $CO_2$ yr$^{-1}$) [78]. For the Amazon rainforest, it was proven that forest degradation contributed three times more to the loss of aboveground biomass than deforestation [79]. Apart from that, the exposure of the soil during silvicultural processes (logging or planting) can lead to a higher decomposition of SOM and thus to considerable carbon losses from belowground biomass [54,73]. Moreover, the capacity of forest ecosystems to store carbon might be reduced under climate change conditions that do not enhance forest growth over the long term due to the expected accelerated life-cycles of forests and additionally lead to comparable high losses of carbon pools in below-ground biomass [44,76,80–83]:

a. Firstly, the photosynthetic activity of mature trees is not expected to be further enhanced due to higher atmospheric $CO_2$ concentrations [54,71–73,75] and even the stimulated growth of younger forests goes along with enhanced respiratory fluxes. Thus, large amounts of the additionally sequestered carbon are released through enhanced respiration [22,72,73]. Apart from that, a transition to a period dominated by vapor pressure deficits that significantly restrict tree growth, health and thus their longevity is expected [84]. There are various indications that higher stem productivity of trees in their early growth period leads to an earlier biomass turnover rate and thus a shorter carbon residence time [76,81,84,85].

b. Secondly, extensive droughts already cause significant carbon losses in tropical forests, which in regular (wetter) years function as carbon sinks, but due to missing precipitation, seasonally turn into carbon sources (on the example of the Amazon, see [44]). Generally, small changes in precipitation have shown significant effects on the carbon fluxes between forest ecosystems and the atmosphere [67,86].

c. Thirdly, also in general, it is expected that soils release more carbon into the atmosphere due to a higher microbial activity. This has been proven for temperate latitudes as well as for the tropics, where carbon losses will be particularly high and [82,83] and is expected to increase by up to 55% due to further changing climate conditions [82].

In sum, the sensitivity of forest ecosystems is mainly influenced by any kind of soil disturbance, climate change and hereby induced weather phenomena, next to the expectable earlier tree mortality, which means there are significant uncertainties in predicting the development of the carbon stock potential of forest ecosystems over time. These factors would need to be considered in earth system model (ESM) projections, which are, however, hardly feasible due to high intrinsic uncertainties [12,54,76,81,85,87,88]. This is why, e.g., the shortened life span of trees is hardly considered in the modelling so far and self-reinforcing processes regarding the loss of SOM are difficult to model accurately [76,83]. This demonstrates the difficulty in accurately depicting increased or decreased sink capacities of forest ecosystems and, therefore, has far-reaching consequences for their associated policy instruments (see Section 4).

3.2.2. Afforestation as a Climate Mitigation Technology

Afforestation and reforestation are both associated with planting and/or deliberately seeding trees on land [89]. However, in contrast to reforestation, afforestation implies land-use changes [89] as it includes planting forests on lands that did contain tree cover before [90]. Afforestation should therefore be assessed differently from the reforestation of areas that are still classified as forests, e.g., due to a canopy density higher than 10% [89], meaning that forests are planted on land that had already contained forests [90]. The FAO, however, connects both with planting and/or deliberate seeding activities, only excluding natural forest regeneration processes [89]. Terms such as "global tree restoration potential" [21], therefore usually include both afforestation and reforestation as they equally refer to the planting and/or deliberate seeding of trees [21]. The boundaries between afforestation and reforestation become partially blurred in practice. Generally, planting trees as a climate change mitigation measure is regularly considered to be economically feasible already, with $CO_2$ prices below USD 50/t $CO_2$ (see in detail [91]). In the EU, it is envisaged to plant at least 3 billion additional trees according to the EU's biodiversity strategy (critically [24]). The Bonn Challenge aims to globally restore 150 million hectares of deforested and degraded land by 2020 and 350 million hectares by 2030 based on the concept of forest-landscape restoration [92]. Thus far, however, the challenge suffers from insufficient participation and requires better forest accounting on national levels [21].

Modelling results regarding the potential to sequester carbon globally by the additional planting of trees until 2100 is, however, challenging and varies—due to contrary assumptions—considerably between 176 Gt $CO_2$ [93] and up to 800 Gt $CO_2$ [94]. Bastin et al., (2019) claim that globally, the conversion of 1 billion hectares into forests with a canopy density higher than 10% could sequester approximately 205 Gt $CO_2$ under current climatic conditions [21]. Yet, they state that emission reductions might decline under changing climate conditions and that, in this regard, the model contains substantial uncertainties [21]. According to Veldman et al., the calculated climate effect is overestimated by at least the factor 5, as SOM gains are most probably lower, the albedo effect is inadequately considered, and the afforestation is included in grasslands and savannas rich in biodiversity, where wildfires and omnivores naturally control the forest cover [22]. Therefore, afforestation can pose major threats to biodiversity-rich natural ecosystems [22–25] and can even increase the risk of spreading wildfires [95,96]. Concerning Europe, models of Strandberg and Kjellström reveal that afforestation of all unwooded areas in Europe could result in a cooling of 0.5–3 °C of seasonal mean temperatures, however, mostly with local and—again—hardly exactly predictable effects [97] and without sufficient consideration of natural site conditions.

In any case, modelling results and the potential contribution of afforestation and reforestation to climate change mitigation must be reviewed critically due to the following:

a.  When estimating the climate effect, next to the challenging assessment of the potential carbon sequestration in forest biomass (see Section 4.2.1), surface albedo and evapotranspiration (the sum of evaporation and transpiration) must be considered as interdependent biophysical climatic factors. Forested areas usually have a lower surface albedo compared to unforested areas and conceal the high albedo of snow. This causes a warming effect, which is particularly prevalent in lower latitudes, such as the boreal zone [56,66,97–99]. In contrast, evapotranspiration of forest ecosystems interacts with clouds and influences precipitation so that a cooling effect occurs [66,97,98]. The cooling effect due to enhanced evapotranspiration typically prevails but is particularly pronounced in humid, tropical regions. The extent of these two contradicting effects is therefore determined by the amount of water in the ecosystems, positively influencing the evapotranspiration, and the latitude influencing the planar reflectivity together with the land-use changes, influencing the magnitude of the albedo effect [100]. Therefore, afforestation and reforestation in tropical regions

are estimated to be more effective than in more temperate regions with lower water availability but expectable greater changes in surface albedo [97,98]. In contrast, it is anticipated that afforestation in the boreal zone may even easily lead to adverse climate effects, meaning that it might contribute to global warming [39,101,102].

b.　Apart from the above (a), there might be a limited or even an adverse climate effect of tree planting initiatives caused by reinforcing disturbances under changing climate conditions [56,76,96,99,103–106]. Firstly, increased tree growth requires sufficient water and nutrients such as nitrogen and phosphorus in order to take advantage of rising $CO_2$ content in the atmosphere, which are limited [107–109]. Next to water shortages due to extended droughts, this could be investigated concerning the plant-available phosphorus that becomes further restricted under changing climate conditions, particularly, but not exclusively, in tropical environments [110–112]. Thus, expected enhancements in forest productivity might be considerably constrained by a shortage of essential nutrients such as phosphorus and might not occur in the expected manner. Secondly, with the increasing rising risk of droughts and as a result of the accelerated life cycle of trees, it is highly likely that tree mortality rates will continue to increase globally [87]. Thirdly, as a result of complex biogeochemical processes, the carbon budget of a forest is highly sensitive to any kind of disturbance. Soil disturbances regularly occur in the context of tree planting, converting young forests to conspicuous sources of $CO_2$ [54]. Particularly severe and contradictory climate effects are to be expected when natural carbon reservoirs and biodiversity-rich wetlands or unmanaged grasslands are afforested [4,22,23,113]. In addition to a loss of SOM, natural vegetation gets lost, threatening biodiversity [22,113].

c.　Furthermore, deforestation with successive afforestation might not maintain the same effects on warming and cooling as former old-growth intact forest ecosystems might have done. Despite the fact that forested lands as part of the LULUCF sector in Europe still function as a sink in most of the EU Member States, a declining sink capacity has been recently measured due to increasing demand for timber and biomass for bioenergy as well as natural disturbances [114]. According to the statistics of the FAO, the sink capacity of forested land in 2020 has already declined by nearly 50% compared to 2015 [115]. Naudts et al., (2016) claim in that respect that afforestation and forest management in Europe thus far did not contribute to the mitigation of climate change. Instead, unsustainably managed forests functioned as a net source of carbon [86].

These considerations lead us to a more overarching point (see the following [12] with regard to geoengineering and to the IPCC in general [116]). Discussions about figures and scenarios as such are far less binding for sustainability research than is often assumed. Rather, it is crucial to analyse the background assumptions of various calculations in detail. This is often difficult because sometimes assumptions are not openly revealed or are even completely opaque. In any case, scenarios on potentials are not norms, nor are they forecasts—they are merely projections.

Notwithstanding, assuming favourable natural constraints for tree cover and sustainable forest management, successful tree-planting projects that are evaluated after a longer time span of 50 or even better more than 100 or 200 years, might develop as a net carbon sink, especially if the interacting tree species reflect the natural, potential vegetation and are not regularly disturbed by logging [77,86,117]. Compared to the goal of reaching zero net emissions in less than two decades or even clearly before 2035, this is, however, a long time span and will not substitute for mitigation measures with immediate effect such as phasing out fossil fuel based emissions [12,15,76,113].

Short-term carbon pool gains by afforestation might only be achievable if former agriculturally used, and widely degraded land is managed sustainably and possibly afforested. This is because, especially under intensive arable land use, SOM content tends to decrease, and soil disturbances are regularly higher than under a forest cover [17–19,70,99,105,118]. This leads to another potential conflict associated with large-scale

afforestation: food security. In particular, small-scale farmers could be (further) deprived of their land in the course of afforestation, potentially increasing dependency on food imports that might cause food prices to rise sharply [91,98,119]. Therefore, integrating trees into diversely managed agricultural systems seems to be more convincing than afforest agricultural land on a large scale. This could generate urgently needed resilient food systems that locally contribute to reaching food sovereignty, mitigate climate change and preserve biodiversity [26,39,120]. Agroforestry systems or sowing catch crops to diversify agricultural practices are the starting points [36,39,121]. Agroforestry binds carbon in vegetation and soil through the combination of trees or other woody plants and arable crops or animal husbandry and thus stores more carbon than agriculturally used land without trees [122,123].

In light of the aforementioned considerations, the idea of fighting climate change through planting trees alone must be generally questioned: The effects might be much lower than hoped for or even adverse, as the carbon-sink capacity of young forests and the availability of land are overestimated while land competition and potential trade-offs regarding food security as well as the need for biodiversity protection are underestimated (see [27,56,124]). If reforestation and afforestation are considered climate change mitigation measures by providing negative emission potentials, the manifold ecosystem functions of forest ecosystems and their resilience, next to site-specific natural and socio-economic conditions, require the utmost attention [54,59,60,76,96,125,126]. In other words: The climate mitigation potential of large-scale afforestation, partly overlapping with reforestation, varies widely, particularly in the short term—and is regularly overrated (see also Table 1). Afforestation should only be considered if natural (and cultural) site conditions are favourable and trade-offs regarding biodiversity and food security remain low. This is, however, regularly not taken into sufficient consideration, contrasting human rights and the CBD. The IPCC, therefore, attributes only medium confidence to the climate mitigating effect of afforestation and reforestation measures, in contrast to the high confidence regarding the potential of measures further listed in Table 1.

**Table 1.** Estimated global climate effect of different mitigation options according to assessments of the IPCC (Adapted from [14]).

| Climate Change Mitigation Option (Selection) | Potential (Gt $CO_2e$ $yr^{-1}$) | Confidence |
|---|---|---|
| Forest management 1 | 0.4–2.1 | Medium |
| Reduced deforestation and forest degradation | 0.4–5.8 | High |
| Reforestation and forest restoration | 1.5–10.1 | Medium |
| Afforestation * | 0.5–8.9 | Medium |
| Increased soil organic carbon content | 0.4–8.6 | High |
| Dietary change | 0.7–8.0 | High |
| Reduced food waste | 0.8–4.5 | High |

* Estimates are partly overlapping with reforestation.

The natural scientific data highlights that preserving existing forests and halting not only deforestation but essentially also the degradation of forest ecosystems, as well as their restoration, are more reasonable than large-scale tree planting at any cost. Like this, gains in ecosystem resilience, biological diversity and climate change mitigation as well as adaptation are achieved, and the latter become connected [59,60,127,128]. Furthermore, we have seen that both measurability and the prediction of the carbon storage capacity of forest ecosystems under future climatic conditions will be extremely challenging [84,87]. When trying to depict the additional carbon storage potential, tree-specific and site-specific conditions have to be taken into account, which themselves are influenced by

changing climatic conditions and further anthropogenic interventions [67,86]. Site-specific soil conditions interact with vegetation and precipitation and are highly sensitive, so forest ecosystems might even seasonally change from a carbon sink to a source. A large number of small actors, difficulties in verifying single emission sources, as well as problems with the monitoring occur additionally.

All of this does not only demonstrate that forests are in serious danger of being overestimated regarding their climate protection capabilities. Moreover, the highly heterogeneous empirical findings indicate the same massive governance problem that we call the problem of depicting (see Sections 1 and 2) and that has already played a major role in our earlier contributions to land use in general, biodiversity and especially on peatlands [4,9,56,129]. This must be considered when thinking about optimally designed policy instruments concerning forest governance since, for example, economic instruments need a governance unit that is easy to grasp in order to function well [10,27,130]. Insofar as drivers such as fossil fuels or animal husbandry are addressed, such a unit is available; however, insofar as additional specific rules for forests are to be formulated, this is lacking.

## 4. Results: Status Quo Governance Analysis

Given the background on methodology and the natural scientific review on forests in the climate debate, we apply a qualitative governance analysis of the status quo of the EU forest governance in the following. Although there is no explicit competence for forest policy in the EU primary law, forests are indirectly governed by many EU policies and initiatives, including the biodiversity, agricultural and climate sector [131]. The most important aspects of these policies are analysed in the following.

### 4.1. EU Strategies Related to Forests and Their Management

During the last several years, more and more strategies of the EU recognize forests as important components to achieving various environmental and sustainability targets. The respective strategies are briefly presented in the following. Within the framework of the European Green Deal [132], the Commission points out the need to improve the quality and quantity of forested areas to reach climate neutrality and a healthy environment [132]. To this end, the Commission will, inter alia, take measures to promote imported products and value chains that do not contribute to deforestation and forest degradation [132], which is also in line with the Communication on Stepping up EU Action to Protect and Restore the World's Forests [59] and the Farm to Fork Strategy [133].

Next to the Farm to Fork Strategy (see below), a core element of the Green Deal is the EU Biodiversity Strategy for 2030 [134], aiming at putting biodiversity on the path to recovery by 2030 through protecting and restoring nature in the EU [134]. Primary and old-growth forests are one main focus of the strategy because they are biodiversity-rich ecosystems with high climate value [134]. In line with the CBD, the Biodiversity Strategy intends to ensure a contribution to reverse biodiversity loss [134], which does not only call for the strict protection of remaining forests but also for the restoration of degraded forests as well as re-/afforestation according to specific criteria.

Rather than aiming at strict protection, restoration and re-/afforestation, the EU Forest Strategy from 2013 focuses on sustainable forest management, which is defined as "using forest and forest land in a way, and at a rate, that maintains their biodiversity, productivity, regeneration capacity, vitality and their potential to fulfil, now and in the future, relevant ecological, economic and social functions, at local, national, and global levels, and that does not cause damage to other ecosystems" [58,135]. However, despite this overall definition of sustainable forest management in the Helsinki Declaration elaborated by Forests Europe, a particular interpretation of sustainable forest management varies within the EU Member States and is rather "linked to factors such as the economic importance of the forest sector, forest policy priorities, and the forest ownership structure" [136].

To promote sustainable forest management in Europe and globally, the 2013 EU Forest Strategy already emphasizes the funding of forestry measures, e.g., by the European

Agricultural Fund for Rural Development (EAFRD), the EU's Environment and Climate Action Programme LIFE 2014–2020, the Cohesion Fund and the Solidarity Fund (in the case of major natural disasters such as storms and forest fires), REDD+ and the EU Forest Law Enforcement, Governance and Trade (FLEGT) Action Plan as well as by the research fund Horizon 2020 [58,131]. Nevertheless, the strategy lacks specific objectives, such as the level of funding or a timeframe for the implementation of sustainable forest management practices. Furthermore, the strategy considers forests, above all, as valuable contributors to the bio-based economy. This is also true for the accompanying blueprint for the EU forest-based industries [58], which aims at stimulating growth and increasing the competitiveness of wood-based and related products and services [58,137].

Currently, the Commission has released a new EU forest strategy building on the 2030 Biodiversity Strategy, that "recognises the central and multi-functional role of forests" [138]. The key objectives of the new forest strategy include effective afforestation, forest preservation and restoration in Europe to increase the absorption of $CO_2$, reduce forest fires, promote bio-economy and biodiversity as well as optimise the use of wood in line with the cascading principle—thus to first produce durable wood products, to extend their service life, to re-use them, to recycle them, use them for bioenergy production and only in the last step to dispose of them [139]. Therefore, a revision of the legislation on forest reproductive material shall also take place by 2022. The carbon farming initiative—to be presented in late 2021 [140]—additionally seeks to establish a regulatory framework for certification of carbon removals from tree planting, forest restoration, improved forest management practices and forest biomass production for long-lasting products, including forest managers and owners [138,139]. A respective carbon removal certification should be adopted by the end of 2021.

To enhance the quantity, quality and resilience of forests, the new Forest Strategy includes a roadmap for planting at least three billion additional trees in the EU by 2030 to achieve biodiversity targets and climate neutrality [138,139]. If trees are planted, careful planning with regard to the aim, a multiyear timeframe, the monitoring, the area, the selection of mixed-species, resilient, (native) trees and stakeholder involvement is required [141]. The latter is partly included in the new Forest Strategy, e.g., by allowing only native tree species to be planted unless they are no longer adapted to projected climatic and pedo-hydrological conditions [139].

Afforestation, reforestation and particularly tree planting are to be promoted by the CAP Strategic Plans (see Section 4.4), the Cohesion Policy funds, LIFE programme, Horizon Europe research and innovation funding programmes, the aforementioned carbon farming initiative, as well as further state aid and private sector funding [139]. In this way, payments for ecosystem services and carbon farming (synonymous with carbon sequestration) practices will be rewarded. However, tree planting neither enhances the quality and the resilience of existing forests automatically nor avoids forest dieback. It can even have only minor or even adverse effects on carbon sequestration and biodiversity if valuable, biodiverse treeless ecosystems such as historic grasslands are threatened (see Section 3). It therefore remains to be seen to what extent the voluntary guidelines and criteria set out in the tree planting pledge can successfully prevent potential adverse effects [24,39,141]. Biodiversity-friendly afforestation and reforestation are envisaged to take place according to voluntary guidelines to be established within the closer-to-nature forest management certification scheme [138] by 2023. However, to enhance ecological effects, the EU's focus should be even more on reducing forest degradation through tree harvesting and further disturbances such as road construction through forests. Furthermore, the natural restoration processes of forests, which have hitherto been disregarded, should be supported [24,141]. In that respect, the EU Commission has proposed a legally binding instrument specifying the conditions for ecosystem restoration, focussing on forest ecosystems with a high carbon-storage potential as listed in Annex I of Habitats Directive [142], in late 2021. Apart from that, the definition of primary and old-growth forests should be sharpened in favour of their mapping, monitoring and foreseen strict protection

even by this date. Additionally, until 2023 "thresholds or ranges for sustainable forest management" [138] are intended to be established. However, once again, initially on a voluntary basis together with the closer-to-nature forest management certification scheme [138].

In addition, the Farm to Fork Strategy covers forests at various points. Firstly, the strategy recognizes the interdependence of the increasing frequency of forest fires, new pests, extreme weather and food security [133]. Secondly, the strategy emphasizes the objective to reduce the EU's contribution to global deforestation and forest degradation, which is why the Commission will present a legislative proposal and other measures to avoid or minimize the placing of products associated with deforestation or forest degradation on the EU market soon [134]. At the same time, the strategy points out the need to reduce the dependency on critical feed materials such as soy grown on deforested land, for example, by a transition towards more sustainable livestock farming and promotion of EU-grown plant protein and alternative feed materials [133]. Thirdly, the strategy underlines the importance of eco-schemes within the framework of the new CAP to fund agroforestry and supports a minimum budget for eco-schemes [133]. And fourthly, the strategy proposes green business models such as rewarding carbon sequestration measures undertaken by farmers and foresters by public or private carbon markets or via the Common Agricultural Policy [133]. Finally, the Farm to Fork Strategy draws attention to the issue of critical long-haul transportation for primary agricultural, fishery and also forestry products. A limitation of transportation would enhance the resilience of regional and local food systems and reduce transportation emissions [133]. Taken together, the strategy recognizes the manifold important forest-related aspects, yet these also need to be implemented effectively.

The same is true for the EU Climate Strategy (European long-term vision for a prosperous, modern, competitive and climate neutral economy Strategy) [143], aiming at net-zero GHG emissions by 2050, far exceeding the estimated and required time frame [143]. In the strategy, the need for legislation to maintain and enhance EU forest sinks is pointed out. At the same time, forests are considered to be suppliers of biomass for material and energy usage [143]. Before this background, the strategy highlights the need to foster both roles, e.g., by promoting agroforestry. However, at the same time, it calls for "sustainable intensification of forestry" (and agriculture) [143], which is questionable, especially since a reference to the necessity to strictly protect primary and old-growth forests is missing (see also Section 4.2.3).

Additionally, various other EU Strategies are not only linked to the aforementioned strategies but also touch upon forests, e.g., the EU Bioeconomy Strategy [144] and the EU Circular Economy Action Plan [145]. In line with the above-mentioned Climate Strategy, the Bioeconomy Strategy calls for more sustainable management of forests, as they are important suppliers of biomass. Furthermore, both strategies draw attention to enhanced carbon removal by forests, supported, e.g., by voluntary carbon sequestration projects for forest owners funded by LIFE and by forest protection, afforestation and sustainable forest management [144,145]. However, all these strategies are not legally binding, and compliance with them cannot be sanctioned. Nevertheless, they do provide important starting points for the design of binding regulatory instruments. As regards instrumental measures, the most relevant of the above-mentioned regulations will be analysed in more detail in the following.

*4.2. Renewable Energy Directive II—Impact on Forest Ecosystems*

4.2.1. Status Quo

The overall aim of the amended Renewable Energy Directive II (RED II) [146] from 2018 for the period 2021 until 2030 is the promotion of the use of energy from renewable resources as one goal of the EU's energy framework that (thus far) envisages increasing the share of energy from renewable sources to at least 32% compared to the baseline of

2005. The current directive covers all potential sources of renewable energy and consequently includes also renewable energy from agricultural- and forest-grown biomass. According to Art. 2 para. 24 RED II, biomass is defined as "the biodegradable fraction of products, waste and residues of biological origin from agriculture, including vegetal and animal substances, from forestry and related industries, including fisheries and aquaculture, as well as the biodegradable fraction of waste, including industrial and municipal waste of biological origin."

Thus, next to direct land-use changes due to enhanced logging activities in forests, indirect land-use change (ILUC, as an important example for shifting effects as a typical governance problem) might be fostered if biomass production for energetic purposes is not sufficiently legally constrained, which is examined in the following.

First of all, the classification of woody biomass as renewable energy can be questioned in general. The classification creates public subsidies that counteract subsidies paid under approaches such as the CAP (or, at the international level, the REDD+ system) that aim to prevent forest degradation through increased harvesting or clear-cutting (see Section 4.4). In this respect, the directive assumes climate neutrality of the energetic usage of woody biomass if the sustainability criteria that "apply irrespective of the geographical origin of the biomass" (Art. 29 para. 1) are met. Whereas the previous directive from 2009 [147] did not specify any restrictions or sustainability criteria for biomass-derived from forests, the new directive tries to close this loophole. In the next chapter, the sustainability criteria laid down for agricultural- and forest-grown biomass used for the production of biofuels, bioliquids and biomass fuels will be critically assessed (see in detail [56,148]).

If the biomass fuels are gained from agricultural land, firstly, the biomass shall not be taken from land with a high biodiverse value which includes (a) primary forests and other wooded lands characterized by native species and functioning ecological processes, (b) species-rich and not degraded highly biodiverse forests and other wooded land (according to the assessment in 2008), (c) highly biodiverse grasslands and (d) nature and wildlife protection areas according to domestic or international law (Art. 29 para. 3). The latter also includes multilateral agreements and lists drawn up by NGOs such as the International Union for the Conservation of Nature (IUCN). However, an exception clause is made concerning highly biodiverse forests, weakening the criteria: The biomass fuels can be produced if the evidence is provided that their production did not affect nature conservation purposes (Art. 29 para. 3 lit. b). Secondly, Art. 29 para. 4 excludes the usage of biomass from land with high-carbon stocks, including wetlands, continuously forested areas and lands of more than one-hectare size with trees higher than 5 m and a canopy cover between 10 to 30 percent. For the latter, again an exemption clause is set: evidence can be provided that the carbon stock is not negatively affected by the usage of biomass (Art. 29 para. 4 no. 4 lit. c). Thirdly, raw materials for biofuel production should not be obtained from peatlands unless (once again) evidence is provided that "the cultivation and harvesting (…) does not involve drainage of previously undrained soil" (Art. 29 para. 5). Finally, in the case of the usage of agricultural waste and residues for biofuel production, management plans need to address the impact of agricultural production on soil quality and soil carbon (Art. 29 para. 2).

If the biofuel production is based on forest biomass, firstly, national or sub-national laws in the country of harvesting shall ensure legal and long-term sustainable harvesting, forest regeneration by not exceeding the growth rate of forests, the protection of nature conservation areas and the monitoring of forest areas as well as the enforcement of the legislation have to be implemented (Art. 29 para. 6 lit. a). However, if respective evidence by legal requirements cannot be provided (which is the case in most of the world's countries, including the EU), forest management systems need to ensure the latter (Art. 29 para. 6 lit. b). Secondly, the country needs to be a party to the Paris Agreement, must have submitted the NDCs and established national legislation in accordance with Art. 5 PA focussing on the strengthening of sinks (Art. 29 para. 7 lit. a). Again, if respective national legislation to strengthen sinks is missing, its absence can be compensated by management

systems, ensuring that "carbon stocks and sinks levels in the forest are maintained, or strengthened over the long term" (Art. 29 para. 7 lit. b).

In late 2021, operational guidance to demonstrate compliance with the criteria has been given by the EU Commission [149] and by the end of 2026, an assessment of the effectiveness of the criteria shall be carried out, leading to the potential further amendment of the regulation after 2030 (Art. 29 para. 8, 9). The sustainability criteria described above are further combined in Art. 29 para. 10 with mandatory GHG emissions savings for the use of biofuels, bioliquids and biomass fuels. Depending on the time the installation started to operate concerning biofuels and biogas in the transport sector, they shall gradually rise from at least 50% for installations that started to operate before 5 October 2015 up to 60% after 5 October 2015, up to 65% after 01.01.2021, and 70% for electricity, heating and cooling after 1 January 2021 and 80% after 1 January 2026. The highly complex calculation follows Art. 31 para. 1 in combination with Annex VI.

### 4.2.2. Critical Assessment of the Sustainability Criteria

An important point of criticism is that the sustainability criteria to avoid indirect land-use changes (ILUC) or a shifting effect and regarding highly biodiverse forests only apply to biomass sourced from agricultural land and not to biomass sourced from forests (on all following points, see [56,148]). Thus, woody biomass gained from primary and highly biodiverse forests can be harvested and sold officially if the new sustainability criteria explicitly for forests are met. Those are, however, still very weak and vague, particularly concerning the not sufficiently specified sustainable management systems and, above all, lack strict biodiversity-protecting regulations [150]. Apart from that, "to minimize the administrative burden", (Recital 104) Art. 29.1 stipulates that the sustainability criteria for both agricultural and forest sourced biomass only apply to electricity and heating from biomass fuels produced in installations with a total rated thermal input equal to or exceeding 20 MW (solid biomass fuels), and with a total rated thermal input equal to or exceeding 2 MW (gaseous biomass fuels). Member States are, however, free to extend the criteria to smaller installations, albeit there is no obligation to do so. This is why in these cases, non-complying biomass can simply be sold to smaller plants and the already weak sustainability tend to be further undermined [150].

Moreover, it is highly questionable whether the shifting effect or ILUC-risk due to agriculturally sourced biomass can be sufficiently limited by the sustainability criteria of Art. 29 combined with the regulations in Art. 26 of the RED II Directive. Art. 26 lays down specific rules for bioliquids and biomass fuels produced from food and feed crops, such as palm oil, soybeans, maize, sugar cane or rapeseed and sunflower. The share of fuels produced from food and feed crops in the final consumption of energy in a Member State is restricted to a maximum of 7% (Art. 26 para. 1). The Member States, however, are free to set a lower limit or caps distinguishing the different sources of biomass production and considering the ILUC-risk of feedstuffs. If a Member State decides to set a lower limit, also the minimum share of 14% for the use of renewable energy in the transport sector, according to Art. 25 para. 1 can be lowered accordingly but by a maximum of 7%. Additionally, Art. 26 para. 2 restricts the share of biofuels gained from high ILUC-risk biomass production, which would lead to the extension of agricultural land into areas with high carbon stocks, such as forests, wetland and peatlands (Recital 81), that needs to be considered as significant (see for the determination of a significant expansion [151]). Low ILUC-risk crops are defined by yield increases through improved agricultural practices and, in general, productivity promoting schemes as well as by their cultivation on land not previously used for the cultivation of crops (Recital 82). For the years 2020 until 2023, the share of biofuels and bioliquids gained from the cultivation of crops with a high ILUC-risk shall not increase level from 2019 and then, from the beginning of 2023 until the end of 2030, gradually decrease to a level of 0%. However, the decision to simply allow biomass to be harvested for energy use from further areas with a proven high ILUC-risk is absolutely

irresponsible in view of the urgent climate and biodiversity crisis. A phase-out only in 2030 is much too late.

Furthermore, impending shifting effects from one crop to the other are not sufficiently considered. This becomes clear taking into account the Delegated Regulation 2019/807 of 13 March 2019 that supplements the RED II Directive in this respect. According to the Annex of Regulation 2019/807, palm oil is considered the only crop with a high ILUC-risk, with a share of 45% of expansion into the continuously forested and wooded area according to Art. 29 para. 4 lit. b and c of RED II and a share of 23% into wetlands according to Art. 29 para. 4 lit. a of RED II. In contrast, soybean has only attributed a share of 8% concerning its potential expansion in forested and wooded areas. However, in reality, it is estimated that additional soy production could take place mainly in Latin America, covering 2.4 up to 4.2 million hectares of additional cropland and thus, "vast evidence about deforestation and land-use change linked to the cultivation of soy" [152] exists [153]. Apart from that, the criteria for low ILUC-risk laid down in the Delegated Act are not strict enough and may lead to a high risk of ILUC "through the back door" [154]. In contrast, advanced biofuels, as listed in Part A of Annex IX (inter alia algae cultivated in ponds or photobioreactors, different kinds of (bio)wastes, used cooking oil etc.) are introduced only very hesitantly. Art. 25 para. 1 RED II foresees a contribution of advanced biofuels and biogas as a share of final consumption energy in the transport sector with at least 0.2% in 2022, 1% in 2025 and 3.5% in 2030 and their energy content may be considered twice in the accounting (Annex IX).

There are some overall aspects of bioenergy that underline how problematic the perspective of RED II is [56,148]. Ideally, bioenergy, like other renewable energies, is climate-neutral; in reality, however, it generates GHGs itself due to processing (and sometimes through its origin, e.g., in rainforest areas). Moreover, biomass provides relatively little energy per plant. It also reinforces the existing problems of conventional agriculture regarding biodiversity loss, soil degradation, water pollution or disturbed nitrogen cycles [27]. In addition, imports from developing countries exacerbate problems with food security. Furthermore, bioenergy for the North, cultivated on high-yield tropical soils, competes with traditional biomass use in the countries of the Global South, for example, as a building material. Nevertheless, bioenergy appears to be attractive since it is always available, unlike wind and solar energy. But this will gradually change [27] via options such as new power lines, storage facilities and power-to-X; furthermore, wind and solar energy are much cheaper options. The current attempt to promote only the kind of bioenergy in the EU which meets certain criteria, i.e., bioenergy not produced in the rainforest, does not promise a truly radical solution, given the above-mentioned governance problems. Firstly, it is almost impossible to verify these EU criteria anywhere in the world when it comes to administrative implementation (enforcement problem). Secondly, there are the above-mentioned shifting problems: The Brazilian bioenergy producer can simply place its bioenergy plants on non-rainforest fields in response to a ban of this kind and instead create other production areas, such as feed for Western meat consumption, all the more in rainforest areas. Thirdly, the many challenges of bioenergy cannot be depicted as criteria on which the admissibility of bioenergy could depend: How does one intend to determine, for example, whether the individual bioenergy plant has endangered the world food situation or not? Fourthly, there is a lack of ambitious criteria, given that bioenergy is far from climate-neutral—and that biomass is only renewable to a limited extent. Considering biomass from forests, it can be stated that burning wood cannot—or only with a few exceptions—be seen as a carbon-neutral process [155,156]. The carbon from the forest stock is transmitted to the atmosphere within minutes and stays there for a long time. To recover the carbon originally saved in the harvested and burned wood will need decades or centuries or might even never be achieved at all. Thus, considering this slow-in-fast-out principle, it becomes clear that the assumed climate neutrality—despite the however insufficient sustainability criteria—is not justified. A different assessment results only in the case that forest biomass from waste and residues is used for energetic purposes. This

is why, in the future, only residues from traditional forestry management (i.e., leftovers after use for timber, board, paper etc.) or naturally fast-decaying wood as a result of forest dieback from diseases or fire with very low payback periods should be fostered as advanced under RED III [156–160].

In contrast to that, it was calculated that more than 100% of Europe's annual harvest of wood would be needed to supply just one-third of the RED II Directive's renewable energy target [161]. This is why under the current directive, even further increases in forest biomass harvesting can be expected in Europe. The rising demand for wood from the bio-economy has already led to a 69% higher biomass loss between 2016 and 2018 compared to the period between 2011 and 2015 and thus also to a significant reduction in carbon sink capacities of Europe's forest ecosystems [162,163]. This is why, besides this, a sharp increase in the demand for soy, causing further deforestation in Latin America, is expected as well [153]. A further complicating circumstance is that particularly woody biomass (biomass pellets) contains less energy than fossil fuels like coal and that the energy used for felling, transportation, drying and pelleting must be accounted for as well [156,164,165]. Already in 1850, EU forests were almost cut down to zero for energy purposes [161] until fossil fuels had substituted forest biomass, which now needs to be substituted. Thus, lessons should be learned from history, and the same mistakes should not be repeated. It, therefore, seems appropriate to redirect more renewable energy production towards solar and wind power [27,56,166,167] and to strictly limit, but by no means continue to promote, any further use of woody biomass that is not based on the recycling of waste at the end of the life cycle of a product. As far as alternatives like wind and solar power are also resource-intensive and not always free of negative side-effects [27,56,168–170], the implementation of frugality also concerning energy purposes needs to be pursued in parallel. This applies not least to the transport sector, where the simple replacement of combustion engines with electric motors cannot be a solution; instead, completely new transport concepts must be developed, in a renunciation of the overemphasis on individual transport [171,172], for actual research needs see [173].

In combination with the accounting rules for land-use change under the Paris Climate Agreement, which refer to the most up-to-date IPCC guidelines [174], the described effects will be even more severe. The assumption is that the loss of forest biomass is already accounted for in the LULUCF sector of the country of origin. However, this is not necessarily the case due to weak accounting rules, especially if, for example, policy changes can be incorporated in a business-as-usual scenario. This means that the imported forest biomass used in a plant is accounted for as zero emissions in the importing country. In this way, the importation of biomass use for energy production is stimulated, while the responsibility for reporting is shifted to the export countries, which mostly lack effective monitoring and enforcement capacities [156,165].

### 4.2.3. Legal Proposal to Amend the Renewable Energy Directive (RED III)

At the moment, expectations are high that RED III would close existing loopholes in favour of the restoration and protection of forest ecosystems with high biodiversity and carbon value. In particular, there were frequent calls to abandon or at least restrict the promotion of burning biomass generated from forestry and agriculture [175]. These expectations cannot be met by the actual legal proposal of 2021 (RED III proposal [176]), that first of all envisages enhancing the share of energy from renewable resources in 2030 to at least 40% (Art. 3 No. 1 RED III proposal). This is convincing as such—although not ambitious enough with regard to Art. 2 para. 1 PA—but needs supplementary rules that focus on the renewables of wind and solar energy. The absence of such rules brings about the danger of further increasing the demand for bioenergy from forestry- and agriculturally-derived biomass in Europe, a demand that already today cannot be met from agricultural production and timber harvesting in the EU. At the same time, an earlier phasing out of fuels from palm or soy oil was not intended, and ILUC risks or shifting effects were not reassessed—again at the expense of global forest cover.

The most important further changes envisaged in the RED III proposal can be summarized and evaluated as follows: No subsidies will be granted for the use of sawlogs, veneer logs, stumps and roots to produce energy, and from 31 December 2026 onwards, there will be no financial support for electricity from forest biomass produced in electricity-only installations (Art. 3 lit. a, b RED III proposal). However, the industry already burns mainly wood with low financial but potentially high carbon and biodiversity value in power plants that mostly combine electricity and heat generation, or even in old coal-fired power plants that—following an already ongoing trend in the EU—could in the future be completely converted to burning forest biomass instead of coal, with as yet uncertain, but probably enormous detrimental consequences for global forest conservation [177–181]. Additionally, if a region is "identified in a territorial just transition plan" (Art. 3 lit. b ii RED III proposal), this requirement does not apply, and support can still be gained even if the power plant produces only electricity, which fosters the potentially disastrous substitution of coal by woody biomass in coal-dependent regions further.

Thus, neither a general phasing-out of the promotion of the energetic use of woody biomass is envisaged according to the RED III proposal, nor a concentration on the exclusive use of residual materials, e.g., from sawmills or the collection of fine woody debris up to a certain locally defined limit, as also recently proposed by the European Commission's Joint Research Centre (JRC) [182]. Instead, a delegated act on how to apply the cascading principle is to be adopted one year at the latest after the amended regulation comes into force (with hitherto uncertain provisions, Art. 3 RED III proposal) and the sustainability criteria of Art. 29 will be further adjusted as follows: First of all, the sustainability criteria of Art. 29 should apply to all installations producing electricity, heating or cooling related to a thermal input to or exceeding 5 MW and no longer 20 MW, which means that more installations will have to follow the sustainability criteria. Secondly, a ban on the procurement of biomass for energy production from primary forests, peatlands and wetlands is proposed so that the existing RED II no-go areas for agricultural biomass production, according to Art. 29 No. 3–5 will finally also apply to forests. This is, first of all, to be welcomed in order to preserve the last primary forests and peat-and wetland with enormous significance for climate protection. In this way, woody biomass from plantations established on former natural forest land shall be excluded from any potential support by RED III and the conversion of biodiverse natural forests into fast growing plantations be prevented in the future (for this suggestion, see also [183]). However, considering that primary forests are very rare in Europe and—like peatlands and wetlands—should be protected anyway (and partly already are), the criteria still remain insufficient, as all other carbon-rich forest types can still be used for energy without restrictions that go beyond the only slightly adjusted sustainability criteria. As has already been pointed out, to prevent problems such as sufficient control in the global value chain, there should be a clear rejection of the promotion of energy recovery from woody biomass that is not based on residual or waste materials that cannot be further recycled anyway. It remains to be seen how the final version of RED III and the Delegated Act on the cascading principle will ultimately be designed.

### 4.3. Timber Regulation and FLEGT

The EU Timber Regulation (EUTR) No 995/2010 [184] is a product-related regulation that refers to more sustainable forest management and acknowledges that the elimination of illegal logging and related trade cannot be achieved by the EU Member States individually. Rather, the regulation recognises that the EU is an importer of commodities associated with significant deforestation, including crops, feedstuffs and livestock products, which makes a policy important to aim at stopping deforestation and illegal harvesting, not only in the EU, but also abroad [185].

The Regulation is a key component of the EU Forest Law Enforcement, Governance and Trade Action Plan (FLEGT) [186], see also [187] and obliges operators who place timber and timber products on the market to minimise the risk of importing illegally

harvested timber by due diligence [59]. The due diligence system comprises information, risk assessment and risk mitigation. This means the operator must have access to information about the timber and timber products, including the country/region of harvest, species, quantity, details of the supplier and information on compliance with national legislation. In addition, an assessment of the risk of illegal timber in the supply chain of the operator and measures to mitigate this risk, e.g., by additional information and verification from the supplier, are required (Art. 5 and 6 EUTR).

The EUTR applies to imported as well as domestically produced timber and timber products to be placed on the internal market. The EUTR complements and strengthens the FLEGT Voluntary Partnership Agreements (VPA) between the EU and timber-producing countries. These FLEGT VPA create legally binding obligations for the parties to implement a licensing scheme and to regulate trade in timber and timber products (recitals 7 and 8 EUTR). The licensing scheme for imports of timber into the internal market is established in Regulation (EC) No 2173/2005 [188], which lists timber products to which the licensing scheme applies in Annexes II and III and partner countries in Annex I. Building on FLEGT, Art. 3 para. 1, EUTR considers timber embedded in timber products listed in Annexes II and III to the Regulation (EC) No 2173/2005, which originate in partner countries listed in Annex I and which comply with Regulation No 2173/2005, as legally harvested. The same is true for timber of species listed in Annex A, B or C to Regulation (EC) No 338/97 (Art. 3 para. 2 EUTR).

Member States are obliged to lay down rules on penalties for infringements of the provisions of the Regulation, including fines, seizure of the timber and timber products or immediate suspension of authorisation to trade (Art. 19 EUTR). Illegally harvested timber and timber products should not necessarily be destroyed. Instead, it may be used for purposes of public interest (recital 27 EUTR). However, the implementation of these penalties in the Member States varies. Sanctions range from administrative sanctions to criminal prosecution [189]. Altogether, the EUTR lacks a cohesive understanding, application and enforcement throughout the Member States, which narrows its effectiveness [189].

Furthermore, the EUTR does not establish sustainably forest rules itself but aims at procedural standards and improving supply chain transparency. Although Recital 2 of the EUTR recognizes the deficiencies of the institutional and governance framework in a number of timber-producing countries with regard to combating illegal logging and the associated trade, the EUTR fails to address this issue: To be exported to EU countries, the wood must be harvested legally, which means, according to Art. 2 (f) "harvested in accordance with the applicable legislation in the country of harvest", no matter whether the host states harvesting rules are sustainable or not [190]. As a consequence, the EUTR suffers from a weak governance effect in timber-producing countries without strict forest legislation [190]; instead, it manifests the status quo in these countries. The implementation of a definition of locally harvested and/or sustainable forest practices independent of the host countries' legislation into the EUTR would have a stronger governance effect. One approach is to incorporate the CBD principles and targets in the EUTR [190]. However, the EU has not yet made any efforts in this regard. Additionally, it remains to be seen how the envisaged regulation on due diligence throughout the value chain relates to the EUTR and to what extent binding standards will be implemented and enforced to ensure more sustainable forestry practices. It also remains open to what extent key drivers of deforestation, such as the production of animal food, will be addressed.

Thus far, the EU fitness check on the EUTR and FLEGT Regulation [191] revealed that the steering effect of both regulations remains comparatively low. The impact of the EUTR on the volume of timber imports from high-risk sources was considered not to be significant, and the interpretation of 'negligible risk' according to Art. 6 para. 2 lit. c was proven to be subjective, while the stringency of enforcement measures generally varies widely. Thus, illegally logged timber could, at best, be kept out of the EU market but not halted globally [191]. Apart from that, FLEGT mainly suffered from a very slow implementation process and involved a limited number of countries. Only 3% of timber product

imports into the EU were covered by FLEGT licences in 2018 [191]. Finally, the lack of political will, the absence of a robust administration, and corruption were named as factors that generally hinder implementation processes [191].

*4.4. Common Agricultural Policy*

The Common Agricultural Policy (CAP) is the main source of EU funds for forests (for an overall critical view on the CAP, see [192]). The CAP consists of two pillars, of which the first pillar mainly comprises direct payments to farmers and the second pillar covers rural development programs. Around 90% of EU funding for forestry measures comes from the European Agricultural Fund for Rural Development (EAFRD), which is co-financed by Member States (European Commission 2015, p. 14), i.e., Pillar II (governed by Regulation (EU) No 1305/2013 [193]. Art. 21 et seq. of the Regulation (EU) No 1305/2013 lays down forest-related measures for investments in forest area development and improvement of the viability of forests. Support under this measure is granted for afforestation and the creation of woodland (Art. 22), the establishment of agroforestry systems (Art. 23) as well as the prevention and restoration of damage to forests and from forest fires, natural disasters and catastrophic events, such as pest and disease outbreaks, and climate-related threats (Art. 24). Furthermore, investments in improving the resilience and environmental value of forest ecosystems, including ecosystem services of forests such as climate change mitigation (Art. 25), are supported. The same applies to investments in forestry technology and in the processing, mobilising and marketing of forest products, including soil- and resource-friendly harvesting machinery and practices (Art. 26). In addition, Art. 34 Regulation (EU) No 1305/2013 includes payments to forest-environmental and climate services and for forest conservation commitments beyond the relevant mandatory requirements in relevant national legislation. As such, agri-environment climate payments may also be granted for forest-related commitments on a voluntary basis (Art. 28). Apart from this, payments for Nature 2000 and the Water Framework Directive (Art. 30) can be provided to forest holders in order to compensate for additional costs for measures taken to implement the Natura 2000 Directives and the Water Framework Directive. Other measures may also include forest-related commitments, e.g., support for cooperation measures, which may also be granted for co-operative drawing-up of forest management plans or equivalent instruments (Art. 35 para. 2 lit. j Regulation (EU) No 1305/2013).

Within the framework of the CAP beyond 2020, according to the European Green Deal, Member States are required to emphasize forest issues more strongly when designing their national strategic plans and thus incentivising more sustainable forest management and avoiding forest degradation [132]. To this end and in accordance with the objectives of the new Biodiversity Strategy and the Farm to Fork Strategy, Member States shall provide an adequate budget for sustainable practices such as agroforestry, forest restoration and re-/afforestation and for bringing back at least 10% of agricultural areas with high-diversity landscape features such as non-productive trees [134].

However, at present and in the future, the design of the rural development programmes depends on the Member States, and so does the decision on the budget to be provided for measures aiming at more sustainable forest-related practices [131]. At the same time, administrative burdens hinder the implementation of such measures [30]. Moreover, only a small part of the total CAP budget is earmarked for the second pillar (24.4%), i.e., rural development, including agri-environment-climate commitments—and even less for forestry measures. The second pillar thus suffers from chronic underfunding, although it contributes to climate and biodiversity protection. The first pillar instead receives 75.6% of the CAP budget—despite criticism towards its direct payments for their detrimental environmental effects ([194], critically pars pro toto [192,195,196]). Furthermore, the CAP does not prevent the great demand for feedstuff for animal husbandry, which triggers deforestation not only in the EU but also in third countries that export feed (on the challenges of livestock farming, see Section 2). This, and in particular, the weak

financing of the second pillar does not appear likely to change in the future CAP (for the reform proposals, see COM (2018) 392 final, 393 final and 394 final; for a critical review, see [192]). Indeed, on 23 November 2021, the EU Parliament approved the amendments of the CAP compromise package [197] after the super-trilogue from 24 to 25 June 2021. Hence, the new CAP regulations are expected to come into force on 1 January 2023 [198].

*4.5. LULUCF Regulation, Legal Proposals on Due Diligence and Forest Information System for Europe*

The present chapter provides some short remarks on further policy instruments that cannot be analysed in detail for reasons of space in the present contribution. The EU Emissions Trading Scheme (ETS) in its current version (only) covers $CO_2$ emissions from power and heat generation, energy-intensive industry sectors, including the production of mineral fertilisers (the EU ETS as a possible instrument for a fast phasing-out of fossil fuels and an ambitious reduction of livestock products—as drivers of deforestation—will be discussed in Section 5). However, non-ETS emissions occurring from industrial energy supply (heating) and product use as well as from the transport, building, waste and agricultural sector [199] are currently subject to the ESR, which therefore also includes non-$CO_2$ emissions, e.g., nitrous oxide ($N_2O$) from the application of fertilisers on crop- or grassland or methane ($CH_4$) from ruminant enteric fermentation or rice pads, which are converted into $CO_2$ equivalents ($CO_{2equ}$) for the accounting requirements. The Effort Sharing Regulation (ESR) divides the 29% target among the Member States according to their gross domestic products for the period from 2021 to 2030 (Decision 406/2009/EC). The July 2021 proposals aim to tighten the reduction targets for ETS and ESR and to make fossil fuels (including buildings and transport) more subject to the ETS overall. This is to be combined with social compensation and a border adjustment, which goes in a similar direction and will be discussed further (in Section 5). However, at present, it is still completely unclear how the discussion on the EU Commission's proposals will develop.

The LULUCF Regulation (analysed in more detail in [200]) is the third pillar of EU climate policy, which is the focus of the present section. It presents an overall framework for guiding the Member States toward more ambitious measures in terms of climate protection, with no policy instrument directly addressing the citizens. The regulation applies from January 2021 onward. It was adopted in 2018 as a "major step forward in establishing a holistic climate policy for Europe" [201] and is "rife with complexity" [201]. It includes the emissions of $CO_2$, $CH_4$ and $N_2O$ as well as their removals through land management, forests and biomass, with forestry being of particular relevance. For the first time, emissions from forest-based bioenergy are included in the carbon accounting alongside all other forest-related emissions. The heart of the regulation is the no debit rule, requiring net-zero emissions from the sector (Art. 4), meaning that all emissions originating in the LULUCF sector have to be fully offset by the removal of GHG emissions in sinks. From 2021 to 2025, no less than −225 Mt $CO_{2equ}$ of annual net removals shall be generated by the sector. Notably, LULUCF, according to its actual status quo, does not cover all land-use-based emissions but excludes major factors, especially most aspects of livestock farming and land-use-related fossil fuel use, that are partly covered by the ESR (see also Section 1; in detail [9]).

However, the scope for the Member States with regard to the accounting rules, especially for managed forest land, remains high and therefore lacks cohesiveness among them. Apart from that, it is possible to offset the emission reductions achieved against the emissions generated between the Member States. Furthermore, the no-debit rule is softened by various flexibilities on the one hand and is still not sufficient to achieve climate neutrality in the sense of the Paris Agreement on the other. For this to occur, the no-debit rule would have to be extended to all three pillars of the EU climate regime and adapted to the requirements of the Paris Agreement. The latter is envisaged according to the LULUCF proposal in the future, although the intended timetable might be too slow to successfully limit global warming to only 1.5 degrees. Moreover, the problem of depicting is

not solved yet, neither in general nor in particular regarding forestry, which might be improved in the future through the planned increased monitoring and less complex accounting rules. According to a new legislative proposal of July 2021, only minor, non-substantive changes are foreseen in the first compliance period until 2025. However, in the second accounting phase, from 2026 to 2030, the net removal target shall be increased from the current −268 Mt $CO_{2equ}$ annually to −310 Mt $CO_{2equ}$ as a legally binding EU-wide target. This comes relatively late, measured against the urgency of the climate crisis. As the LULUCF Regulation and its new drafts are highly controversial, they will not be discussed further here.

EU legislation contains further references to forests-related issues in Council Directive 1999/105/EC [202] on the marketing of forest reproductive material since the restocking of forests, and new afforestation require high-quality, genetically diverse and site-adapted reproductive material (Recitals 2 and 3 Directive 1999/105/EC). In addition, Council Directive 2000/29/EC [203] establishes protective measures against the introduction of organisms that are harmful to plants or plant products into the Member States from the other Member States or third countries and furthermore aims to prevent harmful organisms from spreading to forests [131].

In addition, Member States' criminal laws and other legislation such as legal acts regarding stolen goods may be applicable in some cases of illegal logging, which may enable the criminal prosecution of operators dealing with illegally harvested timber in the Member States. In addition, Member States may apply the measures established in the OECD Action Statement on Combating Bribery (such as the refusal to grant credit), since illegal logging operators are often involved in bribery and corruption [186].

Furthermore, deforestation-free supply chains may be encouraged through various measures. While public funds should only be granted if they do not conflict with sustainability objectives, private investments can also be linked more closely to sustainability criteria. For instance, investors can demand increased transparency along the investment chain from companies. Such measures are supported by the Shareholder Rights Directive (EU) 2017/828, which amends Directive 2007/36/EC [204]. Transparency measures would also be in line with the Commission's proposal for a regulation on disclosures relating to sustainable investments and sustainability risks and amending Directive (EU) 2016/2341 [205]. In fact, the Non-Financial Reporting Directive [206] already requires large companies to enhance transparency and to disclose non-financial information such as environmental, social and human rights matters ([59]; see also the proposal). Likewise, environmental management and audit schemes, such as EMAS, which are regulated by Regulation (EU) 2017/1505 [207], can help to identify and reduce negative environmental impacts, including deforestation [59].

Furthermore, there are two legal proposals on corporate due diligence regarding deforestation and forest degradation in supply chains. First, there is a proposal to introduce mandatory corporate environmental and human rights (regarding the people in the Global South) due diligence at the EU level, as announced by the European Commissioner for Justice at the end of April 2020 [208], which would be a step towards deforestation-free supply chains. In March 2021, the European Parliament adopted a resolution [209] with recommendations to the EU Commission to prepare and submit a legal proposal concerning a directive on mandatory due diligence and corporate accountability with suggestions for legislation. The legislative proposal will aim at holding companies accountable and liable when they harm human rights, the environment and good governance or contribute to harming them. Connected to this, and acknowledging the EU's contribution to global deforestation, the European Parliament adopted a resolution on deforestation [210] in October 2020 that shall minimize the risk of deforestation and forest degradation associated with products placed on the EU market. In its annex, the resolution contains recommendations to the Commission on an EU legal framework to halt and reserve EU-driven deforestation. Among other things, it states that the "commodities covered by the proposal and their derived products placed on the Union market should not result in, or

derive from, the degradation of natural forests or natural ecosystems due to human activity" (Annex 3.2) and "operators should take all necessary measures to respect and ensure the protection of human rights, natural forests and natural ecosystems (…) throughout their entire supply chain (Annex 4.1; see also [211]). In June 2021, NGOs claimed that the adoption of both proposals was delayed [212]. However, given the negative experiences with the bioenergy sustainability criteria (see Section 4.2.2), it remains an open question whether the intended regulation (instead of clear import bans or border adjustments, as also discussed in Section 5) will represent a substantial step forward or not.

The first proposal for a Forest Risk Commodity Regulation (FRCR) [213] was released in November 2021. The proposal contains binding due diligence obligations for companies that want to place raw materials such as soy, beef, palm oil, wood, cocoa and coffee, as well as products derived from them (such as leather, chocolate and furniture) on the EU market. It must be ensured that the commodities and products concerned do not originate from forest areas that have been deforested or degraded after 31 December 2020. They have to be produced in accordance with the laws of the country of origin. A benchmarking system on deforestation and forest degradation risk, inter alia, shall be used to ensure that only deforestation-free and legal products are allowed on the EU market. This could mark a turning point in the fight against global deforestation emanating from the EU (for initial reactions, see [214]). However, ecosystems such as savannahs and wetlands, which are of great importance for climate protection and biodiversity as well, are not covered by the proposed regulation. The same applies to commodities such as rubber, pork, poultry and maize, so shifting effects are again to be expected. Finally, it should be noted that reliance on the (possibly weak) national laws in the country of origin might weaken the regulation, as was already discussed in the example of the EUTR and FLEGT.

On the consumer side, Regulation (EU) No 1169/2011 [215] obligates producers to provide information on ingredients, including oils of vegetable origin which have to be specified in order to allow consumers to distinguish between various vegetable oils (Art. 18 para. 1 and Annex VII Part A No. 8 Regulation (EU) No 1169/2011). Consumers could, for example, decide to avoid products containing palm oil, which is regularly associated with deforestation. However, such instruments are only of an informational nature. They can be supportive, but they do not replace legally binding standards to effectively reduce deforestation.

Another entry point for sharing forest-related information is the Forest Information System for Europe (FISE), which was launched by the European Commission, in particular DG-ENV, DG-JRC, as well as Eurostat and the European Environmental Agency. The information system provides data on the state and health of Europe's forests, e.g., for policymakers, exports, forest industry and forest owners, forest conservationists and scientists [216]. Such information systems can serve as a basis for decision making regarding the development of effective forest governance. However, given the motivational and governance problem findings, they cannot replace binding measures in terms of economic or command-and-control instruments.

## 5. Discussion and Conclusions: Optimizing Governance Options—And Limitations of the Present Analysis

We have seen that regulatory laws and subsidy laws related to forests in the EU are often inadequate. These instruments insufficiently protect primary and semi-natural forests in Europe. They do not sufficiently curb illegal deforestation in third countries. They do not define bindingly and with legal certainty what can be understood by sustainable forestry, i.e., monocultures/plantations are not excluded. They promote the energetic use of woody biomass, palm oil and soybean oil and thereby direct and indirect deforestation. They do not sufficiently promote recycling and reuse (cascade use) of resources. To the extent that meaningful actions are subsidised, these actions are chronically underfunded at the EU level.

The regulatory law issues can theoretically be eliminated relatively easy. For example, to prevent corruption in some EU countries, special EU authorities could monitor regulatory law in more detail and should be granted corresponding competencies. In contrast, this approach would most likely not work in most developing countries due to lacking institutional structures.

It is questionable whether corrections in regulatory law and subsidy law alone are sufficient. These governance approaches (as seen) are typically not able to effectively solve quantity problems since the conservation and expansion of forests is a quantity problem (as is the protection of climate and biodiversity as a whole). Addressing individual areas, products, or actions typically leads to the governance problems discussed above: enforcement problems, shifting effects, rebound effects, and problems of depicting (only the problem of lacking ambition could, in theory, be solved easily by more ambitious regulations). In previous publications, we demonstrated that these governance problems could be best addressed by economic instruments such as cap-and-trade approaches [27,36,37,130,217]. Policy instruments should—with a view to depictability and enforceability—preferably be based on easy-to-grasp parameters on a broad, substantial and geographical scale to avoid shifting and rebound effects. But as regards forestry, trying to precisely address the GHG and biodiversity relevance of a certain forest takes us once again to the limits of economic instruments in addressing a heterogeneous parameter. The wide range of emissions (and biodiversity decrease) and their precise measurement entail that ambitious cap-and-trade approaches are not suitable as a primary instrument. In that, forestry offers comparable policy challenges like peatland conservation measured against the above-mentioned climate and biodiversity targets [4]. This is remarkable in so far as these cap-and-trade instruments, if they are linked to easily comprehensible control variables or governance units such as fossil fuels or livestock products, can otherwise handle governance problems very well and react to various motivational factors. If, however, a problem of depicting arises and cannot be dealt with by switching to an easily comprehensible control variable, economic instruments reach their limits. Knowledge about the exact relevance of a given (or potential) forest—or even single trees—seems still too fragmentary. This also causes issues with the baseline for calculating the emissions balance.

In contrast to peatland governance, the policy challenge of forests cannot simply be solved by some very ambitious and more or less exemption-free command-and-control obligations. It is pretty obvious that humankind will have to go on using forests in an economical way. Therefore, bans work only for some important areas where any kind of economic activity should be prohibited. The most important option is (once again) to radically address the drivers that cause deforestation and lacking areas for afforestation, namely livestock farming and fossil fuels in various sectors. To this end, earlier publications demonstrated that ETS approaches for fossil fuels and livestock at EU level are highly promising [9,15,27,36,130,217,218]. The EU proposals of July 2021 point in the right direction as they plan to broaden the scope of fossil fuels covered by the EU ETS and intend to strengthen its cap. However, the cap would still be not ambitious enough, loopholes (such as LULUCF-related economic instruments of transnational climate law like the former Clean Development Mechanism or similar economic instruments under Art. 6 PA) would continue to exist, and old certificates would not be erased. Going precisely these steps is what must be done to implement effective quantity governance for fossil fuels. So far, the EU proposals are still not in line with Art. 2 para. 1 PA. Furthermore, there is no proposal for a livestock ETS. Our proposal is as follows (in detail, see [10,27]):

Effective EU sustainability policy is best achieved when, at the same time, a kind of climate club is formed with as many other states as possible taking similar measures and establishing uniform environmental standards. Otherwise, global problems remain unsolvable, and shifting effects will occur. At the same time, border adjustments (see [27]) have to be introduced to target those states that do not participate—again, to avoid shifting effects with ecologically and economically detrimental consequences. Such border adjustments or eco-tariffs create incentives for other countries to join the climate club. In line

with that, in July 2021, the EU Commission proposed to introduce a border adjustment for the EU ETS. The same would have to be enacted for the livestock ETS and a potential pesticide ETS. Compared with civil law regulations, these instruments are a more promising way to establish global supply chains with uniform standards.

In order to achieve environmental goals in agriculture and forestry, quantity governance systems of the kind mentioned have to be supplemented by regulatory and subsidy regulations with certain easily graspable and thus controllable governance units—i.e., little exposed to the typical governance problems. Notably, subsidies cannot replace cap-and-trade approaches addressing the drivers of deforestation (on the following, see [27,219]). Changes in subsidies are inferior to establishing caps and levies, despite some similar effects, since subsidies cannot achieve drastic reductions in terms of fossil fuels and livestock products. Moreover, especially cap-and-trade schemes are more cost-efficient than subsidy schemes since they have a more market-oriented structure. Furthermore, caps and levies have a broader scope than subsidies since they are usually more likely to address, e.g., both the acquisition and the efficient use of products. In addition, social distribution issues do not only arise with caps or levies as subsidies are not free. In forestry, too, subsidy law and regulatory law should therefore focus on individual points where the effect of quantity control systems is not sufficient and where at the same time, the problems of depicting, shifting and enforceability are not expected. In principle, EU regulations are again preferable because of their greater scope, which avoids shifting effects (that come with competitive disadvantages for national economies and can weaken the social acceptance of environmental policy measures).

An ETS for livestock products should be supplemented by a livestock-to-land ratio (no longer for organic farming only), which moderately limits the number of animals per hectare and thus avoids a concentration of the remaining livestock and corresponding regional nutrient surpluses. In doing so, an optimal synergy of climate and biodiversity protection is achieved. If, in contrast, the reduction of livestock numbers was pursued solely by a livestock-to-land ratio, the flexibility of farmers would be low and the costs of the system correspondingly higher [10,37,130].

As a framework, the no-debit rule in the LULUCF sector should also be tightened to set negative emissions as a target. In fact, the ongoing amendment process of the regulation addresses this topic—however, over a presumably (too) long period of time (the concrete level takes us back to the debates on targets and potentials; see Sections 1 and 3).

Another regulatory approach that could be implemented relatively quickly is unconditional and comprehensive protection of natural and old-growth forests in developed countries under nature conservation law, especially in the EU. These forests sequester the most carbon and contain the greatest biodiversity. Protection could be achieved by establishing protected areas with strict prohibitions and controls. To avoid corruption, special EU authorities could monitor the process and should be given appropriate competencies. Likewise, a total drainage ban on peatlands in the EU is useful, combined with a requirement to rewet most peatland sites (except in, e.g., populated areas), as the (former) peatland locations are known, and enforcement would be relatively easy.

Furthermore, the use of bioenergy should be restricted or limited to residues. Exceptions could be made for individual flowering plants [56]; conversely, it seems essential for biodiversity that a large part of deadwood remains in the forest. To these ends, an import ban on energetic biomass and a complete end to domestic bioenergy subsidies are useful. All these regulatory approaches are relatively easy to handle and do not suffer from problems of depicting and enforceability. This could replace the sustainability criteria regime in its current form, which suffers from well-known governance problems of regulatory and subsidy instruments. Alternatively, a moderate increase in general levies on land use would be conceivable [27,56]. An open question is whether, in addition to the regulation of livestock farming and bioenergy, further import bans to, e.g., protect rainforests are necessary and legally feasible under global trade law.

The previous proposals do not replace concrete instruments for the restoration of forest ecosystems and reforestation, which should be oriented towards mixed forests. To this end, subsidies appear necessary. In the EU, these subsidies could be combined with a reform of the CAP. For a sustainable bioeconomy, subsidies should only be provided for public services as a supplement to the instruments already presented. For example, subsidies could target farmers and foresters by remunerating forestry and nature conservation measures.

For developing countries, "standards in exchange for money" could be applied by including such countries in the ETS approaches addressing the drivers of deforestation and providing those countries with the revenues of the system to address specified purposes such as afforestation. In theory, Payments for Ecosystem Services such as REDD+ offer financial incentives for landowners to enhance the environmental performance of the land by allocating a financial value to certain ecosystem services (e.g., carbon sequestration or protection of biodiversity) [220]. Certain improvements to the system could be discussed. Clear tenure rights are important to allocate money to the responsible unit, and effective administrative structures are important to enable enforcement and avoid corruption [221]. Transaction costs need to be minimized to achieve high participation [220]. Wang and Wolf [222] find that there are important co-benefits from PES schemes. Because ecosystem degradation frequently affects marginalized communities and people, PES schemes can provide a financial income to these people while at the same time conserving the ecosystem services they rely on. Illegal logging and hunting can also be prevented if the underlying driver (poverty) is addressed. However, the overall situation remains highly ambivalent. On the one hand, a monetary transfer to the Global South is clearly required. On the other hand, shifting effects due to production replacements (to a forest area that is not included in a PES system) can hardly be avoided—one of the reasons why sustainability criteria for bioenergy failed [221]. However, the problem is likely to be partly addressed by other proposed measures, including especially the livestock ETS combined with border adjustments, import ban for bioenergy and fossil fuel phasing out.

These measures discussed above will trigger not only technical innovations but also frugality. This is generally true for quantity governance instruments but particularly important for forests. The described quantity governance systems reduce the pressure of use on forests. This is especially important for the plastics discourse because fossil-fuel based plastic products can frequently be replaced by woody or agriculturally grown biomass products. However, this replacement seems justifiable only if the introduced instruments initially reduce the pressure of direct and indirect land-use changes at the expense of forests. In addition, certain products—such as disposable plates and cutlery, regardless of the material—could be banned altogether, combined with import bans, as these are easily enforceable regulations. Above all, bioplastics should be required to be fully recycled or biodegradable in the natural environment and not only under laboratory conditions, and better protected against harmful effects with regard to microplastics (see in detail [223]).

The present contribution has attempted to resolve some aspects of sustainable (land-use) governance more precisely than previous work—especially the problem of depicting climate and biodiversity effects in highly heterogeneous landscapes and the major role of addressing fossil fuels and livestock farming as damaging factors for finding integrated solutions of various environmental challenges.

Of course, our study is also subject to limitations. In terms of the status quo, we have only considered the EU policy level. However, since the problems identified (see above) are of a general nature, there is much to suggest that the governance options proposed could also be applicable in other countries. Another limitation is that we have not examined in every detail under which conditions regulatory law is a suitable complement to economic instruments. In terms of forests, e.g., the role of rangers and hunters could be a relevant topic for the enforcement of regulatory law since their performance may be important for the protection of forests (see in detail [224]). Generally instructive are also studies—even if they do not belong directly to governance research but rather to

environmental history—that shed light on the often ideologically charged role of forests in political discourse. In Germany, in particular, such an ideological role has a long tradition, especially in the conservative and reactionary political spectrum (see in detail [200]).

In sum, we have seen that forest governance requires governance options that follow a comprehensive approach, not only addressing forests. If done correctly, forest protection, reforestation and afforestation can offer valuable ecosystem services such as carbon sequestration, biodiversity and climate protection, as well as sustainable livelihoods for people. The possibilities of forests to mitigate climate change are significant but limited. This makes forest (protection) instruments important but not a substitution for a rapid decline in fossil fuel use and livestock farming. In any case, sustainability research can learn a lot from analysing forests and their governance. The problem of depicting as well as shifting (or ILUC) effects are the most severe governance issues that call for effective and coherent governance solutions.

**Author Contributions:** Results, J.S. and B.G.; writing—original draft preparation, J.S., B.G. and F.E.; review and editing, K.H. and F.E.; methodology and discussion, F.E.; supervision J.S. and F.E. All authors have read and agreed to the published version of the manuscript.

**Funding:** Open Access funding enabled and organized by Projekt DEAL. The authors and the Research Unit Sustainability and Climate Policy gratefully acknowledge the German Federal Ministry of Education and Research (BMBF) for funding the BonaRes project InnoSoilPhos (No. 031B0509), as well as the Leibniz Association for funding the Leibniz ScienceCampus Phosphorus Research Rostock.

**Institutional Review Board Statement:** Not applicable.

**Informed Consent Statement:** Not applicable.

**Data Availability Statement:** No new data were created or analyzed in this study. Data sharing is not applicable to this article.

**Acknowledgments:** We thank Jutta Wieding, Katharine Heyl and Dean Nixon for proofreading. Furthermore, we thank Jutta Wieding, Anna Bochmann and Sascha Bentke for contributing aspects to earlier versions of our analysis.

**Conflicts of Interest:** The authors declare no conflicts of interest.

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
