# Peer review of "European Forest Governance: Status Quo and Optimising Options with Regard to the Paris Climate Target"

_sustainability, doi:10.3390/su14074365_

Round 1

Reviewer 1 Report

The ms is well-written and it adds new and relevant information on forest management in context of climate-smart cropping systems. The proposed management systems are well-supported by relevant literature and could be used as a guide in forest management outside of Europe.

Reviewer 2 Report

The study is qualitative research, and its aim is to assesse and develop policy instruments for forest governance in the EU. It conducts a qualitative governance analysis of the most important in struments of EU forest policy and presents optimizing policy options, measured against the binding climate and biodiversity targets under international law.

Research hypotheses were clearly stated, and the results were correlated with them.

The methodology was presented accurately.

The language is very clear and readable.

Literature review is adequate to the subject, but the number of cited papers is too excessive. I recommend reducing the number of cited papers because 9 pages of research are only references. The novelty of research should be more clearly presented.

I suggest creating two other sections, respectively Conclusion section Limitations and Outlook section

Please, ensure English one more time.

Reviewer 3 Report

This is one of the best papers that I have recently read in the field of forest governance. Forest-based policies are very well presented and the structure of the paper is very good.

I would suggest the authors to change those bullet points which appear before certain ideas (pages 6-9) and change them into numbers or letters (a, b, c etc .) because bullets are more used in policy works and less in academic papers.

In terms of the section on risk of forest ecosystems it would be good to mention in one sentence a current trend of far-right and nationalist party leaders in some countries in Europe and elsewhere who had an emerging populist discourse by promoting environmentalism for gaining votes (see Conversi Daniele et al, 2021 in journal Environmental Politics on green environmentalism and climate action of left nationalist parties, see also Doiciar Claudia et al, 2021 in journal Geographica Pannonica on the case of the AUR party leaders).  

Finally, authors nicely mentioned the Farm to Fork Strategy which covers forests at various points. The authors can highlight the role of rangers who are important to be mentioned in the protection of forests - see for instance the article of Wang et al on rangers from the forests of southwest China published in journal Forests, 2021. Also, hunters and their tensions with some livestock workers could be mentioned (see O'Brien Thomas et al, 2019, in journal Identities on how the right to access on grazing and food security is important for shepherds.)
